# Assessment of Minimal Residual Disease by Next Generation Sequencing in Peripheral Blood as a Complementary Tool for Personalized Transplant Monitoring in Myeloid Neoplasms

**DOI:** 10.3390/jcm9123818

**Published:** 2020-11-25

**Authors:** Paula Aguirre-Ruiz, Beñat Ariceta, María Cruz Viguria, María Teresa Zudaire, Zuriñe Blasco-Iturri, Patricia Arnedo, Almudena Aguilera-Diaz, Axier Jauregui, Amagoia Mañú, Felipe Prosper, María Carmen Mateos, Marta Fernández-Mercado, María José Larráyoz, Margarita Redondo, María José Calasanz, Iria Vázquez, Eva Bandrés

**Affiliations:** 1Hematological Diseases Laboratory, CIMA LAB Diagnostics, University of Navarra, 31008 Pamplona, Navarra, Spain; paguirreruiz@unav.es (P.A.-R.); bariceta@unav.es (B.A.); zurinebi@gmail.com (Z.B.-I.); amanu@unav.es (A.M.); marfermer@yahoo.es (M.F.-M.); mjlarra@unav.es (M.J.L.); mjcal@unav.es (M.J.C.); 2Navarra Institute for Health Research (IdiSNA), 31008 Pamplona, Navarra, Spain; mc.viguria.alegria@cfnavarra.es (M.C.V.); teresa.zudaire.ripa@navarra.es (M.T.Z.); aadiaz@alumni.unav.es (A.A.-D.); fprosper@unav.es (F.P.); mc.mateos.rodriguez@navarra.es (M.C.M.); am.redondo.izal@navarra.es (M.R.); 3Hematology Department, Complejo Hospitalario de Navarra, 31008 Pamplona, Navarra, Spain; patrizia.arnedo@gmail.com (P.A.); ajaulop@gmail.com (A.J.); 4Advanced Genomics Laboratory, Hemato-Oncology, Center for Applied Medical Research (CIMA), 31008 Pamplona, Navarra, Spain; 5Hematology Department, Clinica Universidad de Navarra (CUN), 31008 Pamplona, Navarra, Spain

**Keywords:** next generation sequencing (NGS), chimerism, myeloid leukemia, hematopoietic stem cell transplant (HSCT), minimal residual disease (MRD)

## Abstract

Patients with myeloid neoplasms who relapsed after allogenic hematopoietic stem cell transplant (HSCT) have poor prognosis. Monitoring of chimerism and specific molecular markers as a surrogate measure of relapse is not always helpful; therefore, improved systems to detect early relapse are needed. We hypothesized that the use of next generation sequencing (NGS) could be a suitable approach for personalized follow-up post-HSCT. To validate our hypothesis, we analyzed by NGS, a retrospective set of peripheral blood (PB) DNA samples previously evaluated by high-sensitive quantitative PCR analysis using insertion/deletion polymorphisms (indel-qPCR) chimerism engraftment. Post-HCST allelic burdens assessed by NGS and chimerism status showed a similar time-course pattern. At time of clinical relapse in 8/12 patients, we detected positive NGS-based minimal residual disease (NGS-MRD). Importantly, in 6/8 patients, we were able to detect NGS-MRD at time points collected prior to clinical relapse. We also confirmed the disappearance of post-HCST allelic burden in non-relapsed patients, indicating true clinical specificity. This study highlights the clinical utility of NGS-based post-HCST monitoring in myeloid neoplasia as a complementary specific analysis to high-sensitive engraftment testing. Overall, NGS-MRD testing in PB is widely applicable for the evaluation of patients following HSCT and highly valuable to personalized early treatment intervention when mixed chimerism is detected.

## 1. Introduction

Allogenic hematopoietic stem cell transplant (HSCT) is a potentially curative treatment in patients with acute myeloid leukemia (AML) and myelodysplastic syndrome (MDS), reducing risk of relapse and improving overall survival [1,2,3]; however, clinical outcomes still vary among patients [4,5,6,7]. Due to the high mortality rate and treatment failures, improved methods of disease status monitoring are clearly needed for patients with myeloid neoplasia following HSCT. Improved surveillance systems may facilitate earlier therapeutic interventions and potentially prevent disease recurrence by tapering immunosuppression, treatment with lymphocyte donor infusion or initiation of anti-neoplastic treatment [8,9]. Standard methodologies to detect clinical relapse in myeloid neoplasms currently include: morphologic assessment of the bone marrow (BM), minimal residual disease (MRD) detection by flow cytometry, cytogenetic or molecular genetic marker detection, and hematopoietic chimerism testing. BM histological analysis has a reduced sensitivity for clinical relapse detection [10]. MRD assessment by flow cytometry for AML and MDS is often complicated due to variable sensitivity of patient-specific marker expression profiles, and can also be subject to inter-assay and inter-operator variability [11]. For chimerism analysis, short tandem repeat (STR) polymerase Chain Reaction (PCR) assays are generally applicable to all HSCT patients, but are limited by a sensitivity threshold of 1–5% [12,13,14]. Newer techniques to analyze chimerism with higher sensitivity (0.01–0.1%) have relatively recently emerged, such as quantitative PCR analysis using insertion/deletion polymorphisms (indel-qPCR) and droplet-digital PCR (ddPCR) [15,16,17]. However, these assays do not specifically detect the presence of disease, but rather they offer a percentage of recipient’s DNA as a surrogate measure for recurrence. This lack of specificity is particularly problematic in chimerism assays, showing high sensitivity, as non-malignant recipient cell lineages may be present in various sample types without representing disease relapse [18]. To maximize sensitivity and specificity, assays such as reverse transcriptase polymerase chain reaction (RT-PCR) may be applied to follow-up specific genetic alterations [19]; however, this is a major limitation in a disease characterized by a striking broad array of different potential oncogenic events across a notable number of genes.

Recently, next generation sequencing (NGS) has been applied to identify clinically relevant variants in AML [20], and persistent allelic burden after chemotherapy has been associated with higher incidence of relapse [21]. Moreover, the presence of genetic variants before HSCT has been associated with higher risk of relapse and shorter overall survival after HSCT [22,23]. Likewise, several studies have demonstrated that the presence of a higher allelic burden at the time of morphologic complete remission is associated with an increased risk of relapse and mortality in AML patients [24,25] and have suggested that the presence of certain genetic variants at morphologic complete remission could be responsible for high risk [26]. Therefore, there has been a great interest to develop high-sensitivity assays to detect any trace of myeloid malignant cells before and after HSCT.

We hypothesized that peripheral blood (PB) serial samples collected for chimerism status monitorization could be useful for NGS analysis, in order to track genetic variants with no additional invasive biopsy procedures. The aim of the present study was to assess the allelic burden in PB using a custom NGS panel alongside measuring the engraftment status using our laboratory’s standard-of-care technique for chimerism engraftment monitoring of post-HSCT patients. With these combined datasets, we intended to establish the value of NGS data during chimerism monitorization and assess their combined capacity for personalized early discrimination of molecular relapse, in order to facilitate earlier therapeutic interventions when mixed chimerism (MC) is detected.

## 2. Experimental Section

### 2.1. Patient Cohorts and Acquisition of Samples

A retrospective study, approved by the DIANA project review board (0011-1411-2017-000028), was designed to assess the utility of NGS-MRD detection after HSCT using PB samples collected for routine clinical engraftment analysis. We selected 20 patients (12 AML, 8 MDS/chronic myelomonocytic leukemia—CMML) with a variety of chimerism profiles and treatment protocols. Briefly, 12 patients had reduced-intensity conditioning regimen (busulfan plus fludarabine) and 8 patients had a myeloablative conditioning regimen (busulfan plus fludarabine or cyclophosphamide); Graft versus Host Disease (GVHD) prophylaxis was performed with a calcineurin inhibitor (cyclosporine or FK506) with methotrexate; T-depletion was performed for unrelated-donor transplantation; and post-HSCT maintenance therapies were not administered until clinical relapse detection (Table 1). Frequency of chimerism monitoring based on high-risk factors presence and clinical grounds was performed by indel-qPCR analysis on 296 PB DNA samples (mean 15 samples per patient; range 7–29). We selected 75 PB samples for NGS analysis (18 diagnosis, 1 post-induction, and 56 post-HSCT: 45 samples had Mixed chimerism (MC) and 11 had complete chimerism (CC) based on chimerism fluctuations and clinical data (Appendix A). Clinical relapse was defined when leukemia blasts were identified by morphological analysis or flow cytometry, or cytogenetic or non-NGS genetic markers were detected. According to these criteria, two groups of patients were studied: patients who relapsed after HSCT (*n* = 12) and patients without relapse at the end of study (*n* = 8). In both groups, we included patients achieving CC at some point during the follow up and patients with MC after HSCT (Appendix A). Two donor samples and 8 paired-bone marrow (BM) samples were also included (4 diagnosis, 4 follow-up).

### 2.2. Indel-qPCR Chimerism Analysis

DNA was isolated from 400 µL of total PB buffy coat using QIAamp DNA Blood Mini Kit (Qiagen, Hilden, Germany) and quantified by Nanodrop Spectrophotometer (Nanodrop Technologies, Wilmington, NC, USA). Baseline donor and recipient DNA were genotyped with the KMR Genotyping Kit (GenDx, Utrecht, The Netherlands) and informative markers were selected (positive in recipient and negative in donor). Chimerism presence was tested by KMR Track Kit (GeneDx), with post-HSCT DNA (150 ng) and pre-HSCT recipient DNA (10 ng), and the chimerism percentages, represented as host-DNA percentages, were determined using the ddCt method according to the manufacturer’s instructions [27]. We defined complete chimerism (CC) as host-DNA percentage inferior to 0.01% and mixed chimerism (MC) as host-DNA percentage above this threshold.

### 2.3. Next Generation Sequencing (NGS)

DNA samples were quantified using Qubit dsDNA BR Assay Kit on a Qubit 3.0 Fluorometer (Life Technologies, Carlsbad, CA, USA), and quality was assessed by DNA genomic kit on a Tape Station 4100 (Agilent Technologies, Santa Clara, CA, USA). Samples at diagnosis and post-HSCT were analyzed with a custom pan-myeloid panel targeting 48 myeloid genes described by Aguilera-Diaz et al. [28]. Libraries were carried out following manufacturer’s instructions, quantified using the Qubit dsDNA HS Assay Kit on a Qubit 3.0 Fluorometer (Life Technologies), and quality was assessed using the D1000 Kit on the 4100 Tape Station (Agilent Technologies); 8 pooled libraries were normalized at 4 nM and pair-end sequenced on a MiSeq Sequencer (Illumina, San Diego, CA, USA) with 251 × 2 cycles using the Reagent Kit V3 600 cycles cartridge (Illumina, San Diego, CA, USA).

### 2.4. Variant Data Analysis

Fastq files were uploaded onto SOPHiA DDM software (SOPHiA GENETICS, Saint Sulpice, Switzerland) for alignment, variant calling, and annotation, filtering out intronic and intergenic variants. Aligned reads were manually curated with the Integrative Genomics Viewer (IGV) software (Broad Institute, Cambridge, MA, USA).

In addition, two in-house hotspot variant calling analyses were performed using VarScan version 2.4.2 [29] and GATK version 4.0.8.1 Mutect2 [30] to detect variants with variant allele frequency (VAF) below 1% threshold. The filtering values for VarScan analysis were: strand bias; minimum coverage: 2; minimum supporting reads at a position to call variants: 2; minimum base quality at a position to count a read: 1; and minimum VAF: 10-5. For Mutect2 analysis, the parameters were: minimum base quality required to consider a base for calling was reduced to 1, the minimum phred-scaled confidence threshold at which variants should be called to 1 and the maximum number of reads to retain per alignment start position was disabled. Mutect2 was run in tumor-only mode and with hotspots as interval list to reduce computing time. Variants from both methods were manually curated to confirm the hotspots selected for each patient.

Clinical classification of the resulting variants was individually reviewed according to the Spanish Group of Myelodysplastic Syndromes guidelines [31]. Post-HSCT monitoring was performed considering all NGS-trackable variants, meaning variants that: (i) were classified as pathogenic, likely pathogenic, or variants of uncertain significance (VUS); (ii) had a minimum coverage of 500 reads; (iii) had a minimum of 12 reads of the alternative allele; and (iv) had a VAF ≥ 0.1% with at least one of their time points with VAF > 5%. Regarding MRD by NGS in post-HSCT, a sample was considered NGS-MRD positive when a variant with clinical relevance, including pathogenic and/or likely pathogenic variants, was detected.

## 3. Results

### 3.1. Assessment of the NGS Sensitivity on PB Samples

First, we assessed the sensitivity of NGS on PB samples in comparison to BM paired samples by Pearson correlation test. We compared 4 PB and BM samples at diagnosis, and similar VAF were detected showing similar sensitivity (R^2^ = 0.9891; *p*-value < 0.0001). Besides, comparison of 4 PB and BM samples at follow-up times showed high correlation (R^2^ = 0.9978; *p*-value < 0.0001) (Appendix A).

These results showed similar sensitivity of NGS on PB and BM samples both for the diagnosis and follow-up, confirming that PB samples are also suitable for molecular testing when BM is not available.

### 3.2. Identification of NGS Variants in PB of Myeloid Neoplasms

We analyzed samples collected at the time of diagnosis (*n* = 18) or at post-induction treatment time (*n* = 1); no sample before HSCT was available for unique patient number (UPN)20. The remaining 19 patients showed a total of 57 variants. Considering variants of UPN20 and de novo acquired variants during the follow-up, the number of total detected variants increased to 63 (mean 3.15 per patient). These variants classified as pathogenic (*n* = 31), likely pathogenic (*n* = 3), and VUS (*n* = 29) showed a broad range of VAF (0.21–88.84%) and were spread across 25 genes. NGS data help to better stratify 3 AML patients shifting from intermediate to high risk group due to the presence of *RUNX1* variants (UPN7, UPN9, UPN11) (Table 2).

To determine the value of molecular NGS-MRD for the discrimination of relapse or non-relapse when MC was detected, only variants classified as pathogenic and likely pathogenic (*n* = 34) were considered (Table 3). The patient without sample before HSCT (UPN20) with a NGS-MRD variant during the follow-up was also included for molecular relapse associated analysis. The NGS-MRD variants were spread across 16 genes (*KRAS*, *TP53*, *DNMT3A*, *FLT3*, *NPM1*, *SRSF2*, *IDH2*, *NRAS*, *PTPN11*, *ASXL1*, *EZH2*, *IDH1*, *PHF6*, *RUNX1*, *TET2*, *U2AF1*), and included 27 single-nucleotide variants(SNV) and 7 indels. The most frequent altered genes were *KRAS* and *TP53* (4 patients), *DNMT3A*, *FLT3*, *NPM1*, and *SRSF2* (3 patients) (Table 2).

We found that 14 patients had 25 variants in clonal hematopoiesis of indeterminate potential (CHIP)-associated genes (*DNMT3A*, *SRSF2*, *CUX1*, *TET2*, *TP53*, *UA2F1*, *ASXL1*, *SF3B1*) [32,33]. Of those, 8 patients harbored more than 1 variant (6 patients with 2 variants, 1 patient with 3 variants, and 1 patient with 4 variants).

These results demonstrate that NGS performed on PB samples is also suitable to characterize the molecular clonal heterogeneity of the myeloid malignancies, and provides useful information to improve the risk stratification of myeloid patients.

### 3.3. Molecular Variants and Chimerism Dynamics after Allogenic HSCT

Low level of host-DNA can be detected in PB for several months after transplant by high-sensitive indel-qPCR assay. Therefore, to determine the presence of molecular markers in the same PB samples would be useful for the interpretation of these low levels of MC. In our study, kinetics of chimerism and genetic variants detected in 56 samples post-HSCT showed a similar time-course pattern (Table 3). Accordingly to chimerism status, 45 samples had MC and 11 had CC. Specifically, in 31/45 (69%) of the samples with MC, we detected NGS-variants; even with MC values below 5% (15 samples). We did not detect any variants in 14/45 of the samples with MC; 8 of those had MC values below 1%. In addition, within the 11/56 samples with CC, 9 samples (82%) showed no molecular variants (Table 3).

These results indicate that NGS might provide additional useful information to chimerism status data during follow-up after-HSCT.

### 3.4. NGS-MRD Specificity in PB Samples from Non-Relapsed Patients

In order to establish the specificity of molecular NGS-MRD in PB, we monitored the pathogenic or likely pathogenic variants of 8 patients in remission with different chimerism status for at least 12 months after HSCT (20 samples).

According to chimerism profile, in 6/8 patients, MC decreased until CC was reached (Figure 1), with a mean time of 220 days (range 90–360 days) (Table 3). During MC time, no NGS-MRD variants were detected in 5/6 patients and in 3/6 patients only VUS was present (Table 3). For UPN16, a pathogenic variant detected during MC time disappeared when CC status was achieved, while a VUS in the NF1 gene (VAF ≈ 50%) confirmed in his sibling-donor was detected at all follow-up samples (Table 2, Appendix A).

In two non-relapsed patients, we detected an increase of MC after HSCT. In UPN8, although MC was persistent and high (>10% host-DNA), no variant was detected at days 90, 180, and 1135 post-HSCT. Surprisingly, for patient UPN9, despite the fact that relapse had never occurred, we detected an increase of VAF for the variants in the CHIP-associated genes *ASXL1* and *SRSF2* concomitant to the MC increase (Table 2, Appendix A). In summary, NGS-MRD was negative at the last time point tested in 7/8 non-relapsed patients, and in 5 of those, the NGS-MRD status totally correlated with CC.

### 3.5. NGS-MRD Sensitivity in PB Samples from Relapsed Patients

To assess the sensitivity of NGS-MRD detection in PB samples during post-HSCT follow-up, we tested 36 samples from the relapsed group (12 patients). In 8 patients (67%), positive NGS-MRD correlated with the presence of MC at the time of clinical relapse (Table 3). All variants detected at relapse were already present at diagnosis; and additionally, in UPN1, two new acquired VUS, not present in his HSC donor, were also identified, suggesting clonal evolution and disease progression (Figure 2). In two patients with CC and negative NGS-MRD (UPN2, UPN3), NGS-MRD was detected when slight increase in chimerism was measured (0.67% and 0.12% host-DNA)(Figure 2 and Appendix A). Importantly, in 6/8 patients, NGS-MRD was detectable between 20 to 220 days (mean 40 days) before clinical relapse (Table 3, Figure 3 and Appendix A).

In 4 relapsed cases, no NGS-MRD was detected: in UPN5 (1.4% host-DNA) and UPN18 (MC > 5%), VUS in *GATA2* and *CUX1* respectively were detected; in UPN4, early relapse was detected with a low MC value (0.2% host-DNA) and was quickly treated; and in UPN11, NGS-MRD was not detected despite the fact the MC value was high (Appendix A).

These results showed that the high specificity of tracking the same NGS variants during HSCT follow up when an increase in MC is detected could help to discriminate early relapse, providing a useful tool for personalized therapeutic intervention.

## 4. Discussion

The present study aims to investigate the clinical value of post-HSCT NGS-MRD monitoring on serial PB samples in patients with myeloid neoplasms according to chimerism status. Clinical decisions after HSCT, such as lymphocyte donor infusion or removal of immunosuppression, are partially based on chimerism results. Considering that MC can have different clinical implications, including disease relapse, graft failure, and rejection, but may also remain stable for a long time and be compatible with prolonged remission [34], identification of patients who could benefit from an early clinical intervention is necessary. We have focused on patients with low levels of MC in hope that close monitoring and NGS-MRD detection could help to take specific clinical decisions such as better timing for the initiation of antineoplastic treatment.

qPCR is as a sensitive method to detect chimerism and previous studies have established cut-off values or increased MC values as a predictive marker for relapse [35,36,37]. In our cohort, NGS provided useful information to understand clinical status during MC fluctuations and the kinetics of early relapse. Our results suggest that the decision of therapeutic intervention in patients with low levels of MC should be based not only in a defined cut-off value, but also in the individualized chimerism kinetics. For instance, NGS could help to discriminate between MC status with positive NGS-MRD (UPN3) and without positive NGS-MRD (UPN7) (Figure 1 and Figure 2).

Moreover, the use of techniques with higher sensitivity and changes in treatment such as reduced intensity conditioning regimens and T-cell depletion [38] have increased the chances to detect the presence of MC. In our cohort, all patients had MC status after HSCT and the time to achieve CC ranged from 90–600 days, considering 0.01% threshold and 70–240 days with a limit of 0.1%. Therefore, chimerism status needs to be comprehensively interpreted and it is desirable to combine it with an additional method that increases specificity. We have showed the NGS utility in 6 non-relapsed patients where MC was not accompanied with NGS-MRD variants, and in one patient where NGS-MRD variants disappeared when CC was achieved (Figure 1). These findings indicate that the disease course is effectively monitored through combination of both techniques and personalized therapy measures can be implemented if needed.

Different studies showed that the presence of allelic burden by NGS at day 21 post-HSCT can estimate the risk of relapse and mortality, and that NGS-MRD monitoring in PB on days 90 and 180 post-HSCT is predictive for relapse and overall survival [39,40]. Our study has demonstrated that monitoring allelic burden by NGS during the disease course is useful to define molecular relapse, and thus could help to take therapeutic decisions. We detected specific positive NGS-MRD in 67% of the patients with relapse and, importantly, in 6 patients, it was detected between 20 to 220 days before clinical relapse (Table 3).

This finding supports similar results showing positive NGS-MRD in 62% of 58 samples (39 patients) collected 20–80 days prior to relapse [41]. Most NGS panels set their sensitivity around 1% of VAF for SNV variants, implying that NGS would not be a suitable technology for MRD detection. However, we found that detecting the same NGS variants present at diagnosis during the follow up after HSCT was useful for clinicians to raise a red flag and keep a closer monitorization.

Besides, personalized chimerism monitoring revealed that a slight increase of MC (<1%) detected by the high-sensitive indel-qPCR method, not detectable with STR-PCR (sensitivity 1–5%), could identify the accurate timing to perform NGS. Recently, simultaneous variant and single nucleotide polymorphism (SNP) based chimerism NGS study in 14 MDS patients detected an increase of MC and variants in 3 patients with relapse [42]. However, SNP-based chimerism sensitivity is lower than with indel-qPCR, and the cost of several serial samples analysis by NGS will be too high to be implemented in the clinical routine. Similarly, simultaneous molecular and chimerism detection by ddPCR has been demonstrated as a suitable approach for disease monitoring post-HSCT in AML [43]. However, ddPCR limits the number of molecular markers that can be assessed, and new clonal variants indicating progression, like the ones found in UPN1, could be missed.

Importantly, 8 patients had variants in CHIP-associated genes [32] at relapse or the last moment of follow up. Nowadays, these variants are difficult to interpret in the context of the disease progression, so further studies are needed to help to discriminate CHIP variants from clonal disease variants. Besides, it has been previously published that clonal hematopoiesis of donor origin cells may be detected [33]. Altogether, we demonstrate that the evaluation of CHIP variants must be done carefully and that the complete genotyping of donors should be implemented.

Importantly, we have used the same PB DNA samples to analyze chimerism and variant status, showed that they perform similarly to BM DNA, and demonstrated the convenience of combining both methods (Table 3). Therefore, the more accessible PB samples could be used to detect MC increase to determine the precise timing to perform NGS, and allow a cost-benefit use of this technique. Overall, we have defined an approach based on NGS-MRD analysis when slight changes of chimerism in PB samples are observed, combining the high-specificity NGS with high-sensitivity chimerism technology.

Despite the advantages of the proposed approach, our patient cohort was limited and therefore we were not able to establish solid values for sensitivity, specificity, and prediction of relapse. Besides, in few relapsed cases, no NGS-MRD was detected, maybe due to the different sensitivity of the technology among variants types (SNV or INDELS) or the fact that some patients may relapse with variants in genes not included in the panel. Therefore, future studies using larger cohorts with serial samples following HSCT would be needed to further confirm the suitability and sensitivity of NGS during chimerism monitoring.

In summary, NGS offers a deeper understanding on variant dynamics throughout the course of post-HSCT and its clinical relevance. Overall, regardless the reason of relapse, the treatment, or the prognosis, this small series shows that personalized NGS-MRD monitoring in combination with highly-sensitive-chimerism analysis are complementary tools to assess early relapse, providing valuable information to monitor myeloid patients after HSCT.

## Figures and Tables

**Figure 1 jcm-09-03818-f001:**
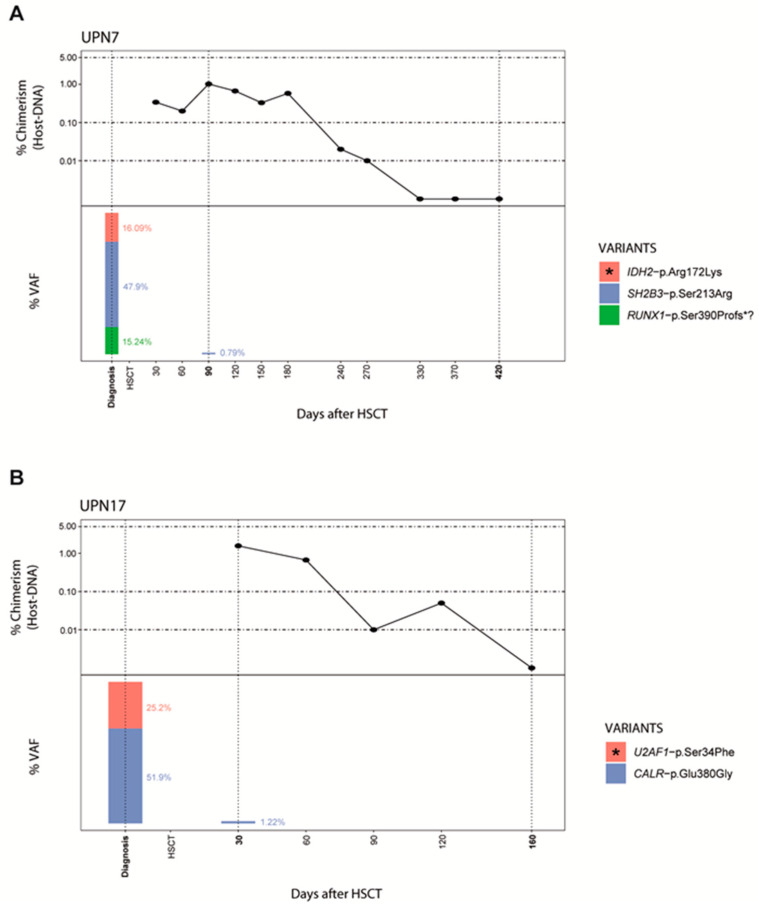
Specificity of the NGS-MRD analysis in non-relapsed patients. Specific negative NGS-MRD confirms remission during MC decreased until CC is reached in both UPN7 (**A**) and UPN17 (**B**). Post-HSCT engraftment analysis by indel-qPCR results are plotted as percentage of receptor (*Y*-axis) over time shown as days post-HSCT (*X*-axis). Vertical dotted lines denote the NGS-analysis time points and the height bars represents VAF percentages; asterisk indicate NGS-MRD variants. (NGS = next generation sequencing; MRD = minimal residual disease; MC = mixed chimerism; CC = complete chimerism; HSCT = hematopoietic stem cell transplant; UPN = unique patient number; VAF = variant allele frequency).

**Figure 2 jcm-09-03818-f002:**
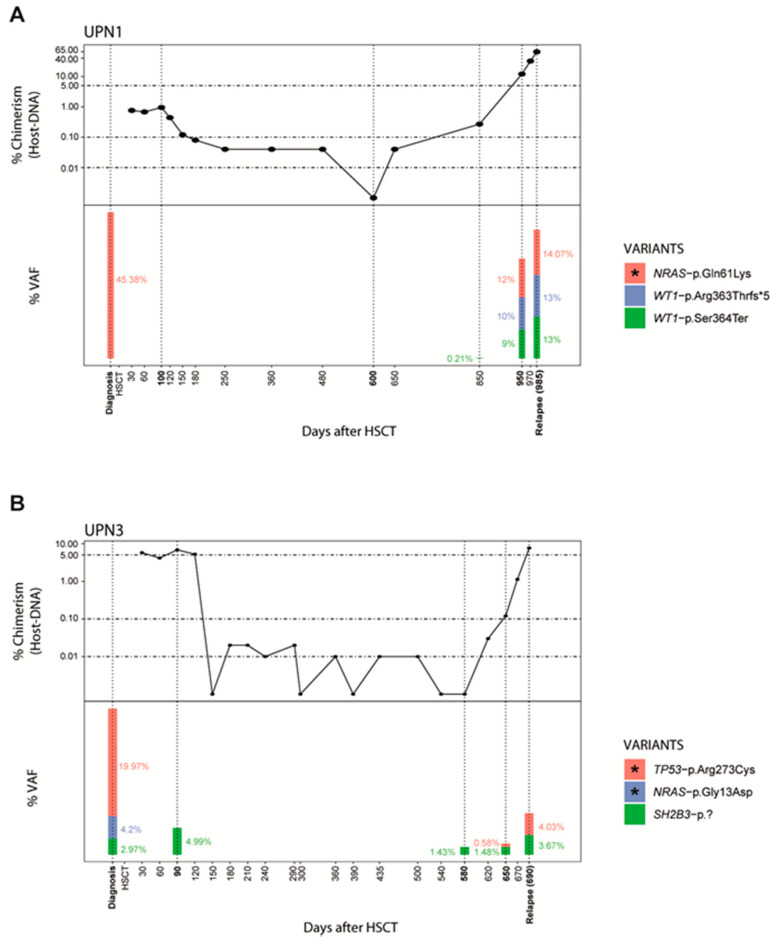
NGS-MRD markers for relapse detection in patients that achieved complete chimerism. Relapsed patients showed a correlation of chimerism status and NGS-MRD during the monitoring of the disease course; MC increase and NGS-MRD variants were detected prior to clinical relapse. (**A**) In UPN1, negative NGS-MRD correlated with CC and two new variants were detected with the slight increase of MC even before positive NGS-MRD presence. (**B**) In UPN3, no complete clearance of all the variants was achieved even during CC, and NGS-MRD turned positive when a slight increase of MC was detected. Post-HSCT engraftment analysis by indel-qPCR results are plotted as percentage of receptor (*Y*-axis) over time shown as days post-HSCT (*X*-axis). Vertical dotted lines denote the NGS-analysis time points and the height bars represents VAF percentages; asterisk indicate NGS-MRD variants. (NGS = next generation sequencing; MRD = minimal residual disease; MC = mixed chimerism; CC = complete chimerism; HSCT = hematopoietic stem cell transplant; UPN = unique patient number; VAF = variant allele frequency).

**Figure 3 jcm-09-03818-f003:**
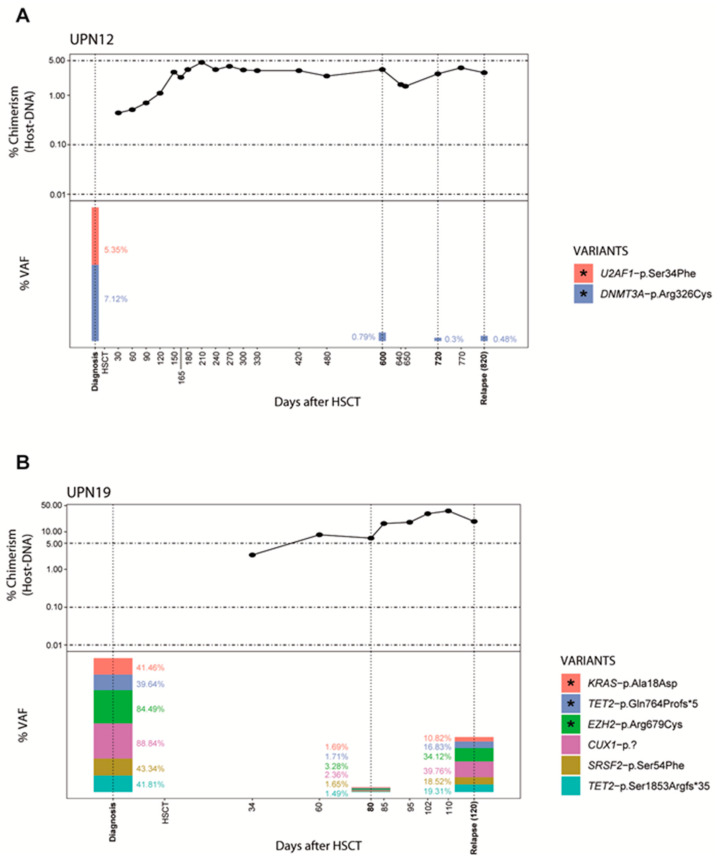
NGS-MRD markers for relapse detection in relapsed patients with MC fluctuations. NGS-MRD during MC monitoring helps to anticipate clinical relapse. Detection of positive NGS-MRD anticipates relapse 220 days in UPN12 (**A**) and 40 days in UPN19 (**B**). Post-HSCT engraftment analysis by indel-qPCR results are plotted as percentage of receptor (*Y*-axis) over time shown as days post-HSCT (*X*-axis). Vertical dotted lines denote the NGS-analysis time points and the height bars represents VAF percentages; asterisk indicate NGS-MRD variants. (NGS = next generation sequencing; MRD = minimal residual disease; MC = mixed chimerism; CC = complete chimerism; HSCT = hematopoietic stem cell transplant; UPN = unique patient number; VAF = variant allele frequency).

**Table 1 jcm-09-03818-t001:** Clinical and therapeutic characteristics of myeloid patients included in this study. Genetic risk was defined by specific scores: ELN for AML, IPPS-R for MDS and CPSS for CMML; pre-transplant disease status was determined by analysis of bone marrow morphology; MRD pre-transplant was determined by flow-cytometry or the presence of a single-molecular marker; and HSCT conditioning regime was selected accordingly to patient fitness.

UPN	Sex	Age at HSCT	Diagnosis	AML/MDS Diagnosis	Genetic Risk	Classical Genetic Markers	NGS Genetic Markers Pre-HSCT	Pre-HSCT Disease Status	MRD Pre-HSCT Status	Days from Diagnosis to HSCT	HSCT Conditioning Regimen	Immunosupression Treatment	HLA Antigen Match	Chimerism Profile after HSCT	Chimerism Profile at Relapse	Clinical Outcome
1	M	18	JMML	de novo	intermediate	46,XY (3 0)	*NRAS*-p.Gln61Lys	CR1	positive	137	MA (BuCy)	FK506 + MTX + ATG	fully matched unrelated donor	CC	MC	relapse
2	F	66	AML	de novo	intermediate	46,XX,t (4;12)(q12;p13)(14)/46,XX(16) *FLT3*-ITD(-)	*IDH2*-p.Arg172Lys *NF1*-p.Ile1603Val *DNMT3A*-p.Val895Met *DNMT3A*-p.Arg729Gln	CR1	positive	188	RIC (FLU + BU2)	FK506 + MTX	fully matched unrelated donor	CC	MC	relapse
3	F	70	AML	Secondary	adverse	46,XX,del(5q)(22/25)/46,XX(3/25)	*TP53*-p.Arg273Cys *NRAS*-p.Gly13Asp *SH2B3*-p.?	Not CR	positive	231	RIC (FLU + BU2)	FK506 + MTX	fully matched sibling donor	CC	MC	relapse
5	F	65	AML	de novo	adverse	hypodiploid complex karyotype	*TP53*-p.Val173Met *GATA2*-p.Gly149Arg	CR1	positive	121	RIC (FLU + BU2)	CS + MTX	fully matched sibling donor	CC	MC	relapse
4	F	61	AML	Secondary	adverse	47,XX,-3,del(5)(q13q33),+8,-17,+21,+21(6)/48,idem,+20(3)/46,XX(7)	*TP53*-p.? *ETV6*-p.Arg291Glyfs*25	CR1	positive	144	RIC (FLU + BU2)	FK506 + MTX + ATG	single antigen mismatch unrelated donor	MC	MC	relapse
11	M	37	AML	de novo	intermediate	46,XY(25)	*PTPN11*-p.Gly503Glu *RUNX1*-p.?	CR1	ND	129	MA (BuCy)	CS + MTX + CAMPATH	fully matched unrelated donor	MC	MC	relapse
12	M	69	MDS	Secondary	adverse	trisomy 8 and monosomy 7	*DNMT3A*-p.Arg326Cys *U2AF1*-p.Ser34Phe	CR1	ND	177	RIC (FLU + BU2)	CS + MTX + CAMPATH	fully matched unrelated donor	MC	MC	relapse
13	F	57	MDS	de novo	adverse	45,XX,-7(4)/45,X,-X(3)/46,XX(13)	*KRAS*-p.Gly12Cys	Not CR	positive	259	MA (FLU + BU4)	FK506 + MTX	fully matched unrelated donor	MC	MC	relapse
14	F	59	AML	de novo	intermediate	47,XX,+4(5/20)/46,XX(15/20)	*FLT3*-p.Val592Ala *NPM1*-p.Trp288Cysfs*12 *DNMT3A*-p.Arg882His *KRAS*-p.Gly12Asp *KMT2A*-p.Gln147Arg	CR1	negative	161	RIC (FLU + BU2)	FK506 + MTX	fully matched sibling donor	MC	MC	relapse
18	F	56	MDS	Secondary	adverse	46,XX,inv(3)(q21q26)(20)	*PHF6*-p.Arg274Ter *SF3B1*-p.Ala708Pro *CUX1*-p.Arg554Gln	CR1	positive	155	RIC (FLU + BU2)	FK506 + MTX	fully matched sibling donor	MC	MC	relapse
19	M	59	CMML	de novo	intermediate	45,X,-Y(1)/46,XY(3)	*KRAS*-p.Ala18Asp *TET2*-p.Gln764Profs*5 *EZH2*-p.Arg679Cys *CUX1*-p.? *SRSF2*-p.Ser54Phe *TET2*-p.Ser1853Argfs*35	Not CR	ND	1750	RIC (FLU + BU2)	FK506 + MTX	fully matched sibling donor	MC	MC	relapse
20	F	62	MDS	de novo	adverse	47,XX,+8(17/20)/46,XX(3/20)	ND	CR1	negative	239	RIC (FLU + BU2)	FK506 + MTX	fully matched sibling donor	MC	MC	relapse
6	F	45	AML	*de novo*	adverse	46,XX(13) *FLT3*-ITD(+)	*FLT3*-ITD-p.Tyr597_Glu611dup *NPM1*-p.Trp288Cysfs*12 *DNMT3A*-p.Leu639Serfs*12	CR1	positive	138	MA (BuCy)	FK506 + MTX + ATG	fully matched unrelated donor	CC	-	remission
7	F	42	AML	de novo	intermediate	46,XX(24/25)/47,XX,+8(1/25]) nuc ish(D8Z2x3)(87/145)	*IDH2*-p.Arg172Lys *SH2B3*-p.Ser213Arg *RUNX1*-p.Ser390Profs*?	CR1	positive	136	MA (BuCy)	FK506 + MTX	fully matched sibling donor	CC	-	remission
10	M	39	AML	de novo	adverse	46,XY,t(3;3)(q21;q26) *FLT3*-ITD(+)	*FLT3*-ITD-p.Asp586_Glu598dup *NPM1*-p.Trp288Cysfs*12 *CUX1*-p.Arg219Gln *GATA2*-p.Gly135Trpfs*50	CR1	negative	170	MA (FLU + BU4)	FK506 + MTX	fully matched sibling donor	CC	-	remission
15	F	61	AML	Secondary	adverse	45,XX,-7(6/20)/46,XX(14/20)	*DNMT3A*-p.Arg882His *IDH1*-p.Arg132Cys *DNMT3A*-p.Phe868Ser	CR1	ND	159	RIC (FLU + BU2)	FK506 + MTX	fully matched sibling donor	CC	-	remission
16	M	39	MDS	de novo	adverse	46,XYY,t(2;11)(q32;q13)?,-5,t(7;16)(q31;q22)?,del(20q)(7)/47,XYY(4)	*TP53*-p.Arg267Trp *RUNX1*-p.Arg139Gln *SRSF2*-p.Pro95Leu *NF1*-p.Leu380Phe	Not CR	positive	262	MA (FLU + BU4)	FK506 + MTX	fully matched sibling donor	CC	-	remission
17	M	41	MDS	de novo	adverse	46,XY,del(12p)(7)/46,XY(18)	*U2AF1*-p.Ser34Phe *CALR*-p.Glu380Gly	Not CR	positive	88	MA (BuCy)	CS + MTX	fully matched sibling donor	CC	-	remission
8	F	56	AML	de novo	adverse	47,XX,+8,t(5;9;11;13)(q33;p22;q23;q13)	*KRAS*-p.Gly13Asp *PTPN11*-p.Ala72Thr	CR1	negative	161	RIC (FLU + BU2)	CS + MTX + CAMPATH	single antigen mismatch unrelated donor	MC	-	remission
9	M	68	AML	de novo	intermediate	46,XY(20)	*ASXL1*-p.Gly646Trpfs*12 *SRSF2*-p.Pro95His *KMT2A*-p.Leu989Phe *NF1*-p.Leu2714Val *RUNX1*-p.Asn82Asp	CR1	ND	162	RIC (FLU + BU2)	CS + MTX + CAMPATH	fully matched unrelated donor	MC	-	remission

UPN = unique patient number; M = male; F = female; AML = acute myeloid leukemia; MDS = myelodysplastic syndrome; JMML = juvenile myelomonocytic leukemia; CMML = chronic myelomonocytic leukemia; MRD = minimal residual disease; ELN = European LeukemiaNet; IPPS-R = Revised International Prognostic Scoring System; CPSS = CMML-specific prognostic scoring system; ND = not determined; CR, complete response; HSCT = hematopoietic stem cell transplant; MA = myeloablative; RIC = reduced intensity conditioning; BuCy = busulfan-cyclophosphamide; FLU = fludarabine; BU2 = busulfan 2 days; BU4 = busulfan 4 days; FK506 = tacrolimus; MTX = methotrexate; ATG = antithymocyte globulin; CS = cyclosporin A; CC = complete chimerism; MC = mixed chimerism.

**Table 2 jcm-09-03818-t002:** NGS variants identified in the 20 patients during the disease time course. Information of the variants detected with the pan-myeloid panel includes VAF percentage and sequencing depth for all time points. For variants with VAF below 1% results from VarScan (SNV) and Mutect2 (indels) in-house analysis are plotted.

UPN	Gene	Chr	Position	Consequence	c.DNA	Protein	Classification	Diagnosis	Post-TM	Post-HSCT 1	Post-HSCT 2	Post-HSCT 3	Post-HSCT 4	Post-HSCT 5	Relapse	Post-Relapse
1	*NRAS*	1	115256530	missense	c.181C > A	p.Gln61Lys	Pathogenic	45.38% 8951x	-	ND	ND	ND	12% 6457x	-	14.07% 5872x	-
	*WT1*	11	32417914	frameshift	c.1086dupA	p.Arg363Thrfs*5	Uncertain significance	ND		ND	ND	ND	10% 7688x		13% 7444x	
	*WT1*	11	32417910	frameshift	c.1077_1090dupGACTCTTGTACGGT	p.Ser364Ter	Uncertain significance	ND		ND	ND	0.21% 6200x	9% 7694x		13% 7400x	
2	*IDH2*	15	90631838	missense	c.515G > A	p.Arg172Lys	Pathogenic	13.91% 4667x	-	ND	ND	-	-	-	0.40% 3716x	1.62% 5002x
	*NF1*	17	29652872	missense	c.4807A > G	p.Ile1603Val	Uncertain significance	48.96% 3619x		ND	ND				0.42% 3352x	1.60% 4634x
	*DNMT3A*	2	25457204	missense	c.2683G > A	p.Val895Met	Uncertain significance	12.61% 5688x		ND	ND				0.47% 4510x	1.81% 5967x
	*DNMT3A*	2	25463307	missense	c.2186G > A	p.Arg729Gln	Uncertain significance	12.04% 6036x		ND	ND				0.37% 4884x	1.41% 6183x
3	*TP53*	17	7577121	missense	c.817C > T	p.Arg273Cys	Pathogenic	19.97% 3445x	-	ND	ND	0.58% 5165x	-	-	4.03% 6688x	-
	*NRAS*	1	115258744	missense	c.38G > A	p.Gly13Asp	Pathogenic	4.20% 4020x		ND	ND	ND			ND	
	*SH2B3*	12	111885351	splice site	c.1236 + 3A > G	p.?	Uncertain significance	2.97% 3810x		4.99% 3810x	1.43% 4186x	1.48% 4987x			3.67% 4792x	
5	*TP53*	17	7578413	missense	c.517G > A	p.Val173Met	Pathogenic	1.32% 7719x	-	0.32% 4999x	ND	-	-	-	ND	-
	*GATA2*	3	128204996	missense	c.445G > A	p.Gly149Arg	Uncertain significance	51.46% 6528x		4.32% 6246x	ND				0.68% 3691x	
4	*TP53*	17	7578370	splice site	c.559 + 1G > A	p.?	Pathogenic	28.94% 7888x	-	ND	ND	-	-	-	ND	ND
	*ETV6*	12	12022762	frameshift	c.870delC	p.Arg291Glyfs*25	Uncertain significance	17.69% 8934x		ND	ND				ND	ND
11	*PTPN11*	12	112926888	missense	c.1508G > A	p.Gly503Glu	Pathogenic	32.91% 5585x	-	ND	-	-	-	-	ND	-
	*RUNX1*	21	36252852	splice site	c.427 + 2T > C	p.?	Uncertain significance	35.04% 1096x		ND					ND	
12	*DNMT3A*	2	25470498	missense	c.976C > T	p.Arg326Cys	Likely pathogenic	7.12% 5648x	-	0.79% 2341x	0.30% 7718x	-	-	-	0.48% 2935x	-
	*U2AF1*	21	44524456	missense	c.101C > T	p.Ser34Phe	Pathogenic	5.35% 5363x		ND	ND				ND	
13	*KRAS*	12	25398285	missense	c.34G > T	p.Gly12Cys	Pathogenic	7.54% 2919x	-	ND	0.58% 1733x	-	-	-	2.38% 3237x	-
14	*FLT3*	13	28608281	missense	c.1775T > C	p.Val592Ala	Pathogenic	23.42% 3151x	-	-	-	-	-	-	ND	ND
	*NPM1 Type A*	5	170837543	frameshift	c.860_863dupTCTG	p.Trp288Cysfs*12	Pathogenic	15.27% 1821x							ND	ND
	*DNMT3A*	2	25457242	missense	c.2645G > A	p.Arg882His	Pathogenic	36.24% 3797x							2.22% 2832x	2.07% 13045x
	*KRAS*	12	25398284	missense	c.35G > A	p.Gly12Asp	Pathogenic	1.94% 2167x							ND	ND
	*KMT2A*	11	118339497	missense	c.440A > G	p.Gln147Arg	Uncertain significance	28.84% 2691x							ND	ND
18	*PHF6*	X	133549136	stop codon	c.820C > T	p.Arg274Ter	Likely pathogenic	12.74% 2834x	-	ND	-	-	-	-	ND	-
	*SF3B1*	2	198266810	missense	c.2122G > C	p.Ala708Pro	Uncertain significance	17.97% 3016x		ND					ND	
	*CUX1*	7	101923357	missense	c.1661G > A	p.Arg554Gln	Uncertain significance	49.19% 3015x		3.27% 6597x					3.31% 2446x	
19	*KRAS*	12	25398266	missense	c.53C > A	p.Ala18Asp	Pathogenic	41.46% 2383x	-	1.69% 5756x	-	-	-	-	10.82% 1303x	-
	*TET2*	4	106157384	frameshift	c.2290dupC	p.Gln764Profs*5	Pathogenic	39.64% 3042x		1.71% 8269x					16.83% 2400x	
	*EZH2*	7	148506462	missense	c.2035C > T	p.Arg679Cys	Likely pathogenic	84.49% 2243x		3.28% 6309x					34.12% 1603x	
	*CUX1*	7	101713618	splice site	c.223-1G > T	p.?	Uncertain significance	88.84% 1945x		2.36% 4997x					39.76% 1484x	
	*SRSF2*	17	74733082	missense	c.161C > T	p.Ser54Phe	Uncertain significance	43.34% 2469x		1.65% 10315x					18.52% 2921x	
	*TET2*	4	106197221	frameshift	c.5557_5558dup	p.Ser1853Argfs*35	Uncertain significance	41.81% 3449x		1.49% 9252x					19.31% 3729x	
20	*SRSF2*	17	74732959	missense	c.284C > G	p.Pro95Arg	Pathogenic	-	-	9.07% 11465x	-	-	-	-	42.72% 11317x	-
	*CUX1*	7	101848405	missense	c.3118G > A	p.Val1040Met	Uncertain significance			15.12% 4187x					42.32% 3852x	
	*TET2*	4	106190851	missense	c.4129T > G	p.Phe1377Val	Uncertain significance			10.52% 6340x					74.32% 5947x	
	*RUNX1*	21	36259163	missense	c.247A > C	p.Lys83Gln	Uncertain significance			1.41% 3757x					5.83% 4271x	
6	*FLT3-ITD*	13	28608223	inframe	c.1788_1832dup	p.Tyr597_Glu611dup	Pathogenic	51% 6880x	-	ND	ND	ND	-	-	-	-
	*NPM1 Type A*	5	170837543	frameshift	c.860_863dupTCTG	p.Trp288Cysfs*12	Pathogenic	36.09% 3497x		ND	ND	ND				
	*DNMT3A*	2	25466788	frameshift	c.1914delT	p.Leu639Serfs*12	Uncertain significance	43.80% 7175x		ND	ND	ND				
7	*IDH2*	15	90631838	missense	c.515G > A	p.Arg172Lys	Pathogenic	16.09% 6232x	-	ND	ND	-	-	-	-	-
	*SH2B3*	12	111856588	missense	c.639C > A	p.Ser213Arg	Uncertain significance	47.90% 5635x		0.79% 2404x	ND					
	*RUNX1*	21	36164626	frameshift	c.1167delC	p.Ser390Profs*?	Uncertain significance	15.24% 4613x		ND	ND					
10	*FLT3-ITD*	13	28608261	inframe	c.1756_1794dup39	p.Asp586_Glu598dup	Pathogenic	43% 4503x	-	ND	ND	-	-	-	-	-
	*NPM1 Type D*	5	170837544	frameshift	c.863_864i-CCTG	p.Trp288Cysfs*12	Pathogenic	36.74% 2730x		ND	ND					
	*CUX1*	7	101758502	missense	c.656G > A	p.Arg219Gln	Uncertain significance	47.41% 3634x		1.19% 2010x	ND					
	*GATA2*	3	128205042	frameshift	c.399_430	p.Gly135Trpfs*50	Uncertain significance	45.04% 4043x		ND	ND					
15	*DNMT3A*	2	25457242	missense	c.2645G > A	p.Arg882His	Pathogenic	10.33% 6246x	-	ND	-	-	-	-	-	-
	*IDH1*	2	209113113	missense	c.394C > T	p.Arg132Cys	Pathogenic	3.82% 5495x		ND						
	*DNMT3A*	2	25457284	missense	c.2603T > C	p.Phe868Ser	Uncertain significance	5.53% 6092x		ND						
16	*TP53*	17	7577139	missense	c.799C > T	p.Arg267Trp	Pathogenic	51.86% 3922x	-	1.72% 7751x	ND	-	-	-	-	-
	*RUNX1*	21	36252865	missense	c.416G > A	p.Arg139Gln	Pathogenic	12.11% 1024x		ND	ND					
	*SRSF2*	17	74732959	missense	c.284C > T	p.Pro95Leu	Pathogenic	5.40% 3539x		ND	ND					
	*NF1*	17	29528130	missense	c.1138C > T	p.Leu380Phe	Uncertain significance	35.46% 2033x		44% 3011x	51% 1413x					
17	*U2AF1*	21	44524456	missense	c.101C > T	p.Ser34Phe	Pathogenic	25.20% 3012x	-	ND	ND	-	-	-	-	-
	*CALR*	19	13054612	missense	c.1139A > G	p.Glu380Gly	Uncertain significance	51.90% 3703x		1.22% 3865x	ND					
8	*KRAS*	12	25398281	missense	c.38G > A	p.Gly13Asp	Pathogenic	38.32% 5128x	-	ND	ND	ND	-	-	-	-
	*PTPN11*	12	112888198	missense	c.214G > A	p.Ala72Thr	Pathogenic	4.63% 6042x		ND	ND	ND				
9	*ASXL1*	20	31022441	frameshift	c.1934dupG	p.Gly646Trpfs*12	Pathogenic	-	1.40% 6069x	1.49% 3293x	1.61% 2231x	1.62% 3769x	7.18% 5675x	16% 14672x	-	-
	*SRSF2*	17	74732959	missense	c.284C > A	p.Pro95His	Pathogenic		1.21%6677x	ND	ND	1.12%3479x	7.11%4879x	16.52%15740x		
	*KMT2A*	11	118344839	missense	c.2965C > T	p.Leu989Phe	Uncertain significance		48.66%6178x	ND	0.69%2036x	1.61%5476x	6.87%4539x	12%4007x		
	*NF1*	17	29687547	missense	c.8140C > G	p.Leu2714Val	Uncertain significance		49.82%5221x	ND	ND	1.87%4547x	5.71%4117x	8.50%3624x		
	*RUNX1*	21	36259166	missense	c.244A > G	p.Asn82Asp	Uncertain significance		0.97%3005x	ND	ND	0.69%2188x	6.27%3143x	11.85%5427x		

UPN = unique patient number; Chr = chromosome; TM = treatment; HSCT = hematopoietic stem cell transplantation; ND = not detected; hyphen (-) = NGS analysis not performed.

**Table 3 jcm-09-03818-t003:** Correlation between chimerism and presence of molecular variants for the 20 HSCT patients. Results show the percentage of chimerism in total peripheral blood and the presence of molecular markers detected by NGS for all time points during the disease course.

UPN	Diagnosis	Patient Group	Moment of Sample	Days after HSCT	% Chimerism	NGS-Trackable Variants ^1^	NGS-MRD Variants ^2^
1	JMML	Relapse	Before HSCT	-	-	Positive	Positive
			Post-HSCT	100	0.95%	Negative	Negative
			Post-HSCT	600	<0.01%	Negative	Negative
			Post-HSCT	850	0.3%	Positive	Negative
			Post-HSCT	950	12%	Positive	Positive
			Relapse	985	64%	Positive	Positive
2	AML	Relapse	Before HSCT	-	-	Positive	Positive
			Post-HSCT	250	<0.01%	Negative	Negative
			Post-HSCT	360	0.09%	Negative	Negative
			Relapse	380	0.67%	Positive	Positive
			Post-Relapse	400	2.24%	Positive	Positive
3	AML	Relapse	Before HSCT	-	-	Positive	Positive
			Post-HSCT	90	6.87%	Positive	Negative
			Post-HSCT	580	<0.01%	Positive	Negative
			Post-HSCT	650	0.12%	Positive	Positive
			Relapse	690	7.7%	Positive	Positive
5	AML	Relapse	Before HSCT	-	-	Positive	Positive
			Post-HSCT	90	6.8%	Positive	Positive
			Post-HSCT	540	<0.01%	Negative	Negative
			Relapse	1350	1.41%	Positive	Negative
4	AML	Relapse	Before HSCT	-	-	Positive	Positive
			Post-HSCT	100	0.1%	Negative	Negative
			Post-HSCT	300	0.12%	Negative	Negative
			Relapse	410	0.2%	Negative	Negative
			Post-Relapse	470	0.34%	Negative	Negative
11	AML	Relapse	Before HSCT	-	-	Positive	Positive
			Post-HSCT	100	19%	Negative	Negative
			Relapse	130	67%	Negative	Negative
12	MDS	Relapse	Before HSCT	-	-	Positive	Positive
			Post-HSCT	600	3.3%	Positive	Positive
			Post-HSCT	720	2.7%	Positive	Positive
			Relapse	820	2.85%	Positive	Positive
13	MDS	Relapse	Before HSCT	-	-	Positive	Positive
			Post-HSCT	45	3.6%	Negative	Negative
			Post-HSCT	80	5.2%	Positive	Positive
			Relapse	100	11.6%	Positive	Positive
14	AML	Relapse	Before HSCT	-	-	Positive	Positive
			Relapse	60	5.5%	Positive	Positive
			Post-Relapse	140	<0.01%	Positive	Positive
18	MDS	Relapse	Before HSCT	-	-	Positive	Positive
			Post-HSCT	90	6.2%	Positive	Negative
			Relapse	180	5.5%	Positive	Negative
19	MDS	Relapse	Before HSCT	-	-	Positive	Positive
			Post-HSCT	80	6.7%	Positive	Positive
			Relapse	120	19%	Positive	Positive
20	MDS	Relapse	Before HSCT	-	-	NA	NA
			Post-HSCT	90	29%	Positive	Positive
			Relapse	130	100%	Positive	Positive
6	AML	Remission	Before HSCT	-	-	Positive	Positive
			Post-HSCT	90	0.02%	Negative	Negative
			Post-HSCT	300	0.01%	Negative	Negative
			Post-HSCT	820	<0.01%	Negative	Negative
7	AML	Remission	Before HSCT	-	-	Positive	Positive
			Post-HSCT	90	1.02%	Positive	Negative
			Post-HSCT	420	<0.01%	Negative	Negative
10	AML	Remission	Before HSCT	-	-	Positive	Positive
			Post-HSCT	110	1.79%	Positive	Negative
			Post-HSCT	170	<0.01%	Negative	Negative
15	AML	Remission	Before HSCT	-	-	Positive	Positive
			Post-HSCT	60	0.85%	Negative	Negative
16	MDS	Remission	Before HSCT	-	-	Positive	Positive
			Post-HSCT	90	3.85%	Positive	Positive
			Post-HSCT	360	<0.01%	Negative	Negative
17	MDS	Remission	Before HSCT	-	-	Positive	Positive
			Post-HSCT	30	1.6%	Positive	Negative
			Post-HSCT	160	<0.01%	Negative	Negative
8	AML	Remission	Before HSCT	-	-	Positive	Positive
			Post-HSCT	90	15.4%	Negative	Negative
			Post-HSCT	200	14.9%	Negative	Negative
			Post-HSCT	1140	33%	Negative	Negative
9	AML	Remission	Before HSCT	-	-	Positive	Positive
			Post-HSCT	100	0.21%	Positive	Positive
			Post-HSCT	370	1.7%	Positive	Positive
			Post-HSCT	1250	2%	Positive	Positive
			Post-HSCT	1360	10%	Positive	Positive
			Post-HSCT	1550	26%	Positive	Positive

^1^ NGS-trackable variants: including variants classified as pathogenic, likely pathogenic, or VUS. ^2^ NGS-MRD variants: including variants classified as pathogenic or likely pathogenic. UPN = unique patient number; HSCT = hematopoietic stem cell transplant; NGS = next generation sequencing; VUS= variant of unknown significance; MRD = minimal residual disease; NA = not available; Hyphen= not performed (Chimerism assay is done after HSCT).

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
