# Peer review of "Assessment of Minimal Residual Disease by Next Generation Sequencing in Peripheral Blood as a Complementary Tool for Personalized Transplant Monitoring in Myeloid Neoplasms"

_jcm, 2020, doi:10.3390/jcm9123818_

Round 1

Reviewer 1 Report

ASSESSMENT OF MINIMAL RESIDUAL DISEASE 2 BY NEXT GENERATION SEQUENCING IN 3 PERIPHERAL BLOOD AS A COMPLEMENTARY 4 TOOL FOR PERSONALIZED TRANSPLANT 5 MONITORING IN MYELOID NEOPLASMS 6

Paula Aguirre-Ruiz1, Beñat Ariceta1,2, María Cruz Viguria2,3, María Teresa Zudaire2,3, Zuriñe 7 Blasco-Iturri1, Patricia Arnedo3, Almudena Aguilera-Diaz2,4, Axier Jauregui3, Amagoia Mañu1,2, 8 Felipe Prosper2,4,5, María Carmen Mateos2,3, Marta Fernández-Mercado1,2,4, María José Larrayoz1,2, 9 Margarita Redondo2,3, María José Calasanz1,2, Iria Vazquez1,2* and Eva Bandres2,3*

Review Comments

The study attempts to assess and monitor minimal residual disease (MRD) by using NGS performed on the peripheral blood sample as a diagnostic tool in patients post HSCT for myeloid neoplasm. There are several ways to detect MRD, such as BM morphology, flow cytometry, cytogenetics/FISH, and chimerism study. However, NGS is the emergent novel sensitive method that has not been well validated. The manuscript is fairly well written.

Critiques:

  1. Line 93-106: flow-chart is necessary to have a better description of how the 12 relapsed patients are selected. It is difficult to tell whether there is a selection bias here. Also, to validate the NGS for post-transplant MRD, more than 30 positive/relapsed cases from a big data pool is recommended. Ideally, the authors should focus on one disease category: either AML or MDS.
  2. Line 103: “Clinical relapse was defined when blasts were identified…”, Here the blasts should be “leukemia blasts or neoplastic blasts” and of note, the post-transplant-related “hematogones=blasts” should be excluded. 
  3. It is unclear whether NGS was performed on the patients at diagnosis, using the same panel as NGS-MRD? We recommended adding two columns in table 1 for pre and post-transplant NGS status. The details will be found in the supplemental table. 
  4. Table 2: regarding % chimerism is not clear: the test was done by peripheral blood by sorting CD3 and CD33 or whole bone marrow. It appeared there were several cases without overt loss of donor cells when analyzed with chimerism, but NGS showed positive for mutations. Sometimes at disease relapse, % chimerism is low, and NGS-MRD is negative. How do the authors explain the finding? 
  5. What is the minimal VAF cut-off for calling positive MRD post-transplant? Are these variants pathogenic or also including categories 4 and 5. The %VAF of variants should be included in table 2 (column for NGS-MRD).

Author Response

Review Comments

The study attempts to assess and monitor minimal residual disease (MRD) by using NGS performed on the peripheral blood sample as a diagnostic tool in patients post HSCT for myeloid neoplasm. There are several ways to detect MRD, such as BM morphology, flow cytometry, cytogenetics/FISH, and chimerism study. However, NGS is the emergent novel sensitive method that has not been well validated. The manuscript is fairly well written.

First, we want to thank you for your time and expertise to review our manuscript and help us to improve and clarify our data. We totally agree that the NGS technology is a sensitive method not well validated for MRD detection and we hope that our study helps to prove the clinical utility of NGS monitoring in myeloid neoplasms post-HSCT as a complementary specific analysis to high-sensitive engraftment testing. Following your remarks, we have rewritten some confusing explanations and discussed suggested ideas.

Critiques:

Line 93-106: flow-chart is necessary to have a better description of how the 12 relapsed patients are selected. It is difficult to tell whether there is a selection bias here. Also, to validate the NGS for post-transplant MRD, more than 30 positive/relapsed cases from a big data pool is recommended. Ideally, the authors should focus on one disease category: either AML or MDS.

This retrospective study was designed to assess the utility of NGS during the follow up after HSCT in a variety of patients for whose chimerism status fluctuations was difficult to interpret both in relapsed and non-relapsed patients. To do so, we tried to select a small cohort representing the reality of those patients. In total we selected 56 samples post-HSCT: 45 samples had MC and 11 had CC. Regarding the relapsed patients (36 samples post-HSCT) we include a sample as close to clinical relapse as possible with MC (except UPN14 that not enough pre-relapse sample was available). And for non-relapsed patients (20 samples post-HSCT) we had 3 patients with MC in all their samples, and 5 patients with CC in the last time point. As suggested by the reviewer, we have prepared a flow chart and submitted it as supplementary Figure S1 (line 102).

We totally agree that a big homogenous data pool is always recommended. However, what we tried to show with this study is that implementation of the NGS analysis in a clinical routine laboratory may be helpful in a wide range of myeloid pathologies and rises a red flag to clinicians in patients with high risk of relapse. We believe that even with this heterogeneous small series we have proven that NGS analysis may be useful and easily implemented in the clinical routine.

Line 103: “Clinical relapse was defined when blasts were identified…”, Here the blasts should be “leukemia blasts or neoplastic blasts” and of note, the post-transplant-related “hematogones=blasts” should be excluded.

We have specified the type of blast in the manuscript.

It is unclear whether NGS was performed on the patients at diagnosis, using the same panel as NGS-MRD?

Yes, the same NGS panel was used for all samples. We have made this remark on the manuscript (line 141).

We recommended adding two columns in table 1 for pre and post-transplant NGS status. The details will be found in the supplemental table.

As suggested by the reviewer, we have been considering including an extra column in Table 1, but since most patients have more than one post-transplant NGS results we believe it might be confusing. So, instead we have decided to change Table S1 within the manuscript as Table 2. With this we hope all the data is easily available and both the VAF and depth are reported for each variant for all follow up moments. In Table 1 we have also made the remark that the genetic markers are found before HSCT.

Table 2: regarding % chimerism is not clear: the test was done by peripheral blood by sorting CD3 and CD33 or whole bone marrow.

In the material and methods section we described that chimerism analysis is performed in total peripheral blood; sorting of CD3 and CD33 was not performed. However, in order to clarify it on Table 2 (now Table 3) we have specified that the chimerism was performed on total peripheral blood (line 128).

It appeared there were several cases without overt loss of donor cells when analyzed with chimerism, but NGS showed positive for mutations.

Yes, we have only case (UPN14) that show CC but with clinical relapse and NGS-MRD. It has been previously published that relapse in the donor cells may happen. In our particular case, the donor and the patient both would be carrying one of the most frequent variants of the one of the most frequent mutated genes in CH. The complete genotyping of donors and implementation of single cell DNA analysis will shed some light into this complex disease progression. We have further discussed this idea on the discussion section of the manuscript.

Sometimes at disease relapse, % chimerism is low, and NGS-MRD is negative. How do the authors explain the finding?

That indeed is a very good question. We have considered different explanations. For instance, the sensitivity of the technology may vary among type of variants (SNV or INDELS) and between regions according to the depth of coverage. And besides, some patients may have a relapse with different variants than at the diagnosis, and although we are using a quite complete panel for myeloid disease it may progress with variants in genes not included. This has been discussed in the manuscript (lines 302-305).

What is the minimal VAF cut-off for calling positive MRD post-transplant? Are these variants pathogenic or also including categories 4 and 5. The %VAF of variants should be included in table 2 (column for NGS-MRD).

We are aware that reporting variants with low VAF it may be risky. However, we wanted to be able to report anything that may have had a clinical relevance for these patients. Therefore, we decided to report pathogenic and likely pathogenic variants which clinical relevance is well known and although we report variants with VAF as low as 0.30% all have at least 12 reads of the alternative allele present before HSCT. We have decided to include this information in the material and methods section too (lines 163-170).

As suggested by the reviewer, we have been considering including the VAF of the variants also in Table 2, but since most patients have more than one variant we believe it might be confusing. So, instead we have decided to change Table S1 within the manuscript as Table 2. With this we hope all the data is easily available and both the VAF and depth are reported for each variant demonstrating that the data is robust.

Reviewer 2 Report

In this manuscript, Aguirre-Ruiz P et al present the clinical utility of Next generation sequencing to determine minimal residual disease after engraftment. The content is not really original. Similar studies were previously published with more patients (for instance Balagopal et al, Plos one 2019). The small number of patients in this cohort is not sufficient to establish solid recommendations. Moreover, this manuscript is sometimes difficult to understand, the figures need to be improved.

 Major points that would need to be addressed:

-In the first paragraph, the comparison between PB and BM samples is made on 4 paired samples. It is clearly not enough to conclude. Moreover, which statistical test was used (Pearson correlation?)? P-value is lacking .

 Minor points that would need to be addressed:

-line 58=> the sensitivity is not 1-5% which would be really low. The good term seems to be the "sensitivity threshold" or "level"

-Figures and tables must be improved. It would have interesting to present a synthesis of different profiles of patients rather than individual plots. Moreover, the manuscript lacks of concordance between the text (line 222, 6 patients) and Figure 1 (2 patients).

-Lots of troubles with punctuations, syntaxis (comma, dot, …)

-Gene names and the term "de novo" should be written in italics.

-This is some repetitions between materiel & methods and results (for instance line 218, explaining what was consider NGS-MRD)

-what is the difference between NGS-trackable variants and NGS-MRD variants? "NGS-trackable variant" term is not enough detailed in the manuscript

-It could be interesting to discuss the case of patient 9

-it could be interesting to speak about Clonal Hematopoiesis of Indeterminate potential

Author Response

Comments and Suggestions for Authors

In this manuscript, Aguirre-Ruiz P et al present the clinical utility of Next generation sequencing to determine minimal residual disease after engraftment. The content is not really original. Similar studies were previously published with more patients (for instance Balagopal et al, Plos one 2019). The small number of patients in this cohort is not sufficient to establish solid recommendations. Moreover, this manuscript is sometimes difficult to understand, the figures need to be improved.

Thank you very much for the time and expertise to review our manuscript, and all remarks and suggestions. We have reviewed the manuscript to rewrite confusing paragraphs and removed duplications, and made an effort to improve English expressions and correct syntax’s errors.

We fully agree that our study has been done in a small series of patients. However, we believe that this series, although heterogeneous, it has been well characterize with 296 samples of PB analyzed by indel-qPCR and 75 of those by NGS. Importantly, we demonstrated that PB is as useful as BM during the follow up of these patients and allows NGS analysis of the same sample when fluctuations in chimerism are observed and risk of relapse is possible. Regarding the publication by Balagopal et al., we have read the interesting study with 46 samples (28 BM and 18 PB) and similar results were obtained. They were able to detect patient-specific mutational evidence for MRD in 62% of all samples collected 20–80 days prior to relapse. However, due to the lower sensitivity of the STR-PCR they did not find correlation between NGS-MRD and chimerism status. This is an important difference with our study in which indel-qPCR technology allows higher sensitivity and therefore higher correlation is found with NGS-MRD. Besides, in our study we could anticipate the clinical relapse up to 220 days.

Major points that would need to be addressed:

-In the first paragraph, the comparison between PB and BM samples is made on 4 paired samples. It is clearly not enough to conclude. Moreover, which statistical test was used (Pearson correlation?)? P-value is lacking.

Thank you for the remark; we missed to write down the test used. The comparison between PB and BM was done by Pearson correlation test and p-values are p<<0,001 in both cases. This information has been included in the manuscript (lines 174-177).

Minor points that would need to be addressed:

-line 58=> the sensitivity is not 1-5% which would be really low. The good term seems to be the "sensitivity threshold" or "level"

We fully agreed and have made the change in the draft.

-Figures and tables must be improved. It would have interesting to present a synthesis of different profiles of patients rather than individual plots. Moreover, the manuscript lacks of concordance between the text (line 222, 6 patients) and Figure 1 (2 patients).

Thank you for the suggestion. We have improved the Tables and Table S1 has been included as Table 2 in the manuscript to simplify the reading of the manuscript. Regarding the figures, as suggested by the reviewer we considered to prepare a Figure with the synthesis of the different profiles, but our small series is quite heterogeneous and that single Figures for each patient better describe the moments where chimerism and NGS analysis have been performed. However, we have add a new Figure S1 better describing the selection of patients and samples. We have reviewed the manuscript and corrected the lack of concordance.

-Lots of troubles with punctuations, syntaxis (comma, dot, …)

We have reviewed the manuscript for syntax errors and we hope it is now easier to read.

-Gene names and the term "de novo" should be written in italics.

Thank you for the comment. We have made the changes on the manuscript (Now Table 2)

-This is some repetitions between materiel & methods and results (for instance line 218, explaining what was consider NGS-MRD)

This repeated explanation has been removed and we have reviewed the document and removed duplications.

-what is the difference between NGS-trackable variants and NGS-MRD variants? "NGS-trackable variant" term is not enough detailed in the manuscript

In this study we have reported variants fulfilling certain quality metrics and classified as pathogenic, likely pathogenic or variants of uncertain significance (VUS), what we defined as NGS-trackable variants. However, regarding the identification of the MRD by NGS, we wanted to report as positives only those cases more likely to have a variant with clinical relevance, therefore for NGS-MRD we excluded the VUS, variants which significance is not yet known.

We have rewrite material and methods in order to clarify this (lines: 162-167)

-It could be interesting to discuss the case of patient 9

-it could be interesting to speak about Clonal Hematopoiesis of Indeterminate potential

Thank you for both suggestions. We find the CHIP biology topic very fascinating and have followed your advice and discussed in the manuscript (lines: 206-209, 358-363).